# In-Line Registered Milk Fat-to-Protein Ratio for the Assessment of Metabolic Status in Dairy Cows

**DOI:** 10.3390/ani13203293

**Published:** 2023-10-21

**Authors:** Ramūnas Antanaitis, Karina Džermeikaitė, Vytautas Januškevičius, Ieva Šimonytė, Walter Baumgartner

**Affiliations:** 1Large Animal Clinic, Veterinary Academy, Lithuanian University of Health Sciences, Tilžės Str. 18, LT-47181 Kaunas, Lithuania; karina.dzermeikaite@lsmu.lt; 2Brolis Sensor Technology, Molėtų Str. 73, LT-14259 Vilnius, Lithuania; vytautas.januskevicius@brolis-sensor.com (V.J.); ieva.simonyte@brolis-sensor.com (I.Š.); 3University Clinic for Ruminants, University of Veterinary Medicine, Veterinaerplatz 1, A-1210 Vienna, Austria; walter.baumgartner@vetmeduni.ac.at

**Keywords:** precision dairy farming, milk composition, biomarker, dairy cattle

## Abstract

**Simple Summary:**

In this study, our goal was to develop a simple-to-use method for evaluating the health status of dairy cows by analyzing the milk composition data collected via in-line sensors. Cows were divided into the following groups: subclinical ketosis (n = 62), subclinical acidosis (n = 14), and healthy group (H; n = 20). Our focus was on quantifying the levels of fat and protein in cow milk and uncovering intriguing patterns. Specifically, cows struggling with a particular health issue (subclinical ketosis) displayed an elevated fat-to-protein ratio in their milk. Conversely, cows with a different health complication (subclinical acidosis) exhibited a reduced fat-to-protein ratio. We scrutinized their blood samples and unveiled correlations between specific blood constituents and variations in milk composition. Notably, heightened concentrations of certain substances in the bloodstream coincided with alterations in milk composition. This insight suggests the potential utility of in-line milk composition as a non-invasive method for assessing the metabolic well-being of dairy cows, circumventing the need for intrusive diagnostic procedures. These findings hold promise for enhancing livestock management practices and promoting animal welfare.

**Abstract:**

This study endeavors to ascertain alterations in the in-line registered milk fat-to-protein ratio as a potential indicator for evaluating the metabolic status of dairy cows. Over the study period, farm visits occurred biweekly on consistent days, during which milk composition (specifically fat and protein) was measured using a BROLIS HerdLine in-line milk analyzer (Brolis Sensor Technology, Vilnius, Lithuania). Clinical examinations were performed at the same time as the farm visits. Blood was drawn into anticoagulant-free evacuated tubes to measure the activities of GGT and AST and albumin concentrations. NEFA levels were assessed using a wet chemistry analyzer. Using the MediSense and FreeStyle Optium H systems, blood samples from the ear were used to measure the levels of BHBA and glucose in plasma. Daily blood samples were collected for BHBA concentration assessment. All samples were procured during the clinical evaluations. The cows were categorized into distinct groups: subclinical ketosis (SCK; n = 62), exhibiting elevated milk F/P ratios without concurrent clinical signs of other post-calving diseases; subclinical acidosis (SCA; n = 14), characterized by low F/P ratios (<1.2), severe diarrhea, and nondigestive food remnants in feces, while being free of other post-calving ailments; and a healthy group (H; n = 20), comprising cows with no clinical indications of illness and an average milk F/P ratio of 1.2. The milk fat-to-protein ratios were notably higher in SCK cows, averaging 1.66 (±0.29; *p* < 0.01), compared to SCA cows (0.93 ± 0.1; *p* < 0.01) and healthy cows (1.22). A 36% increase in milk fat-to-protein ratio was observed in SCK cows, while SCA cows displayed a 23.77% decrease. Significant differences emerged in AST activity, with SCA cows presenting a 26.66% elevation (*p* < 0.05) compared to healthy cows. Moreover, SCK cows exhibited a 40.38% higher NEFA concentration (*p* < 0.001). A positive correlation was identified between blood BHBA and NEFA levels (r = 0.321, *p* < 0.01), as well as a negative association between BHBA and glucose concentrations (r = −0.330, *p* < 0.01). Notably, AST displayed a robust positive correlation with GGT (r = 0.623, *p* < 0.01). In light of these findings, this study posits that milk fat-to-protein ratio comparisons could serve as a non-invasive indicator of metabolic health in cows. The connections between milk characteristics and blood biochemical markers of lipolysis and ketogenesis suggest that these markers can be used to check the metabolic status of dairy cows on a regular basis.

## 1. Introduction

Milk is a preferred matrix for monitoring the health status of dairy cows since it is non-invasive and easy to collect, and it is frequently used to identify ketosis and other production problems [1,2]. Milk fat and β-hydroxybutyrate (BHBA) levels vary, indicating lipomobilization and ketogenesis [3]. Milk comprises fat, protein, lactose, enzymes, vitamins, and minerals as a result of numerous metabolic activities in mammary secretory cells. Various factors, such as breed, feeding habits, ambient conditions, and udder health, have an impact on its composition [4]. Negative energy balance (NEB) can cause health difficulties (e.g., fertility issues and infections) [5]. However, health problems (e.g., digestive or locomotive disorders) might be a trigger for NEB and have a negative impact on NEB in early lactation cows [6]. Traditionally, metabolic state is measured using BHBA or nonesterified fatty acid (NEFA) threshold levels [7]. In early lactation, dairy cows have low plasma glucose concentrations with concomitantly higher levels of NEFA and BHBA [8].

Milk parameters are derived from blood and food components, and understanding the correlations between these characteristics in food, blood, and milk helps determine animal health and production status [4,9]. Potential measurements include milk fat and protein percentages, as well as milk fat-to-protein (F/P) ratio [10]. A milk F/P ratio greater than 1.38 increases the likelihood of a cow becoming clinically ketotic by 2.1 times. A milk F/P ratio greater than 1.5 increases the risk of ketosis. Jenkins et al. [11] recently evaluated milk F/P thresholds of over 1.25, over 1.35, over 1.42, over 1.50, over 1.60, and >1.70, reporting sensitivities and specificities of 100 and 49%, 96 and 59%, 92 and 65%, 75 and 78%, 33 and 90%, and 8 and 96%, respectively. They calculated the ideal F/P cutoff as being 1.42 using a receiver operating characteristic curve; the optimal F/P threshold to balance sensitivity and specificity was 1.50. An increased risk of ketosis is accompanied by a greater F/P ratio. This is most likely related to increased BHBA and fatty acid availability for milk fat synthesis [10]. The fact that milk fat and protein contents may be connected to many other parameters besides energy balance in early lactation complicates the accuracy of utilizing F/P ratio as an index of SCK [12]. Milk fat content may not just reflect circulating adipose fatty acid levels [13], and food and rumen health (i.e., acidity) may also have significant influence throughout this period [14].

The protein concentrations in milk fat are the best indicators for the early detection of cows with severe NEB, which is significant for herd management. In previous research, milk biomarkers were found to be more beneficial for severe NEB prediction than blood plasma biomarkers. As a result, milk biomarkers may be useful for determining the energy balance of individual dairy cows [15]. Modern dairy farming frequently entails coerced milk production, which causes metabolic problems in cows. To predict and prevent such problems and related subclinical diseases, the physiological ranges of biochemical indicators in a clinically healthy herd must be established [1]. Dairy producers’ disease detection tools should be inexpensive, noninvasive, and simple to use [16]. Ehret et al. [17] employed metabolic and milk indicators to monitor ketosis in dairy cows using an artificial neural network (ANN) algorithm and found that these indicators had a higher predictive power than genetic indicators.

Milk has a high potential for use as a diagnostic medium because it is acquired non-invasively and on a regular basis through milking, and standard analysis processes have been in place for many years [18]. As a result, the use of early milk samplings, as well as the probable incorporation of new data sources, should be examined further to improve early warning forecasts. Finally, before putting a model into practice, more data from different farms should be collected and assessed to make the model more resilient to variances in management and rations [19].

The aim of this study was to identify changes in in-line registered milk fat-to-protein ratio for the assessment of metabolic status in dairy cows.

## 2. Materials and Methods

### 2.1. Housing Conditions of Study Animals

While conducting this investigation, the provisions of the Lithuanian Law on Animal Welfare and Protection were followed. PK012858 is the study’s approval number. This study was conducted in Lithuania (55.819156, 23.773541) from 2023.07.01 to 2023.07.31. Dairy cows were kept in free-stall barns with ventilation and were fed a total mixed ration (TMR) that was balanced according to their physiological needs throughout the year. The cows were fed at 06:00 and 18:00 every day, with a typical total mixed ration for high-producing, multiparous cows, consisting primarily of 25% corn silage, 5% alfalfa grass hay, 20% grass silage, 15% sugar beet pulp silage, 30% grain concentrate mash, and 5% mineral mixture. This diet was created to meet or exceed the needs of a 500 kg Holstein cow producing 37 kg milk/d. The ration’s chemical composition was as follows: dry matter (DM) 48.8%; neutral detergent fiber (% of DM) 28.2%; acid detergent fiber (% of DM) 19.8%; nonfiber carbs (% of DM) 38.7%; crude protein (% of DM) 15.8%; and net lactation energy (1.6 Mcal/kg). The cows were milked twice a day, at 05:00 and 17:00, using a parlor system. Of the 1160 cows undergoing clinical examination, 320 were selected (second and subsequent lactation from the first 5 to 30 days after calving). The cows’ average body weight was 550 kg ± 45 kg. The average energy-corrected milk yield (4.2% fat, 3.6% protein) per cow per lactation was 12,500 kg.

### 2.2. Experimental Design

Throughout the study period, the farm was visited two times per week on the same days (on Tuesdays and Thursdays) each week. According to the milk fat-to-protein ratio registered using an in-line analyzer (Brolis Sensor Technology, Vilnius, Lithuania), 320 cows (second and subsequent lactation from the first 5 to 30 days after calving) out of 1160 were selected. Clinical examinations were performed on all 320 cows. According to the results of the clinical examination and blood BHBA from the 320 cows for this study, 96 cows were selected: 62 cows were selected to be in the SCK group; 14 cows were in the SCA group; and 34 cows were in the healthy group. The same cows were examined at each visit. Fourteen cows that were assigned to the healthy group at the start of the study became sick during the experiment and were removed from the study. Therefore, the final number in the healthy group was 20.

The cows in the SCK group (n = 62) were classified as having SCK when at least one value of milk F/P ratio was higher than 1.5 and there were no clinical signs of other diseases after calving (metritis, lameness, mastitis, displaced abomasum, or indigestion), with an average rectal temperature of 38.8 °C and rumen motility of five to six times per three minutes [11].

The subclinical acidosis (SCA) (n = 14) group was classified on the basis of the F/P ratio (<1.2) with moderate-to-severe diarrhea and nondigestive food parts in their feces, and there were no clinical signs of other diseases after calving (metritis, lameness, mastitis, displaced abomasum, or indigestion), with an average rectal temperature of 38.8 °C and rumen motility of five to six times per three minutes [20].

The cows in the healthy group (n = 20) were added after calving, and there was no clinical evidence of illness, with the average milk F/P ratio being 1.2. Throughout the study period, the cows’ clinical state was classified as healthy or unwell. During the whole investigation period, all 20 cows in the healthy group were clinically healthy.

### 2.3. Measurements

All cows had their daily milk fat–protein ratios recorded using a BROLIS HerdLine in-line milk analyzer (Brolis Sensor Technology, Vilnius, Lithuania). This system uses a unique GaSb widely tunable external cavity laser-based in-line spectrometer in the 2100–2400 nm spectral band. Milk flow was monitored using the transmission mode continuously during the milking cycle. The obtained molecular absorption spectra were then processed to determine the composition level of the main constituents. During each milking cycle, the analyzer measured the composition of each cow’s milk continuously. This “mini-spectroscope” was mounted on the milking stalls or on a milking robot in the milk line and did not require any additional reagents or maintenance (Figure 1).

During each milking, the analyzer measures the composition of each cow’s milk continuously throughout the milking, every 5 s. Fat, protein, and lactose concentration dynamics are averaged with weights based on milk flow to obtain single values to represent whole milking. Milk analyzer accuracy was evaluated in the Eurofins lab with resulting values of root mean square error of prediction (RMSEP) of 0.21% for fat, 0.19% for protein, and 0.19% for lactose. (Figure 2).

During each farm visit, blood samples from the 320 cows were gathered, and glucose, β-hydroxybutyrate (BHBA), gamma-glutamyltransferase (GGT), aspartate transaminase (AST), albumins, and nonesterified fatty acid (NEFA) concentrations were measured. The concentrations of BHBA and glucose levels were assessed using capillary blood samples obtained from the ear during each clinical examination. Cleaning and puncturing the skin of the left or right ear was the first step in the sampling technique. If the blood volume was insufficient for the measurement, capillary bleeding was induced by gently pressing the ear skin. If the blood volume retrieved was still insufficient for a meaningful measurement, the ear was punctured again. The front edge of the test strip was dipped immediately into the drop of blood after being inserted into the handheld device [21]. The MediSense and FreeStyle Optium H systems (Abbott, Great Britain) were utilized to measure the levels of plasma BHBA and glucose.

At the same time, blood samples from the jugular vein were taken using an evacuated tube with no anticoagulant (BD Vacutainer^®^, Eysin, Switzerland). The blood samples were centrifuged for 10–15 min at 3500 RPM. The samples were delivered to the Lithuanian University of Health Sciences Veterinary Academy’s Large Animal Clinic’s Laboratory of Clinical Tests. Blood serum was tested using a Hitachi 705 analyzer (Hitachi, Tokyo, Japan) and DiaSys reagents (Diagnostic Systems GmbH, Berlin, Germany) to determine the activities of GGT and AST and the concentrations of albumins. The NEFA samples were evaluated using an automated wet chemistry analyzer (Rx Daytona, Randox Laboratories Ltd., London, UK) and Rx Daytona reagents (Randox Laboratories Ltd., London, UK).

Whole-blood BHBA concentrations were used to diagnose hyperketonemia. To determine the greatest BHBA concentration available, samples were taken during farm visits at the same time relative to the time of feeding each week, that is, within 2 to 4 h of a fresh feed delivery [22]. The cows were held in a resting stall or in a headlock during each sampling in order to take a tiny blood sample from the coccygeal vein using a needled syringe.

### 2.4. Statistical Analysis

All statistical analyses were carried out using the SPSS program (SPSS Inc., Chicago, IL, USA IBM Corp., 2017) version 25.0 of IBM SPSS Statistics for Windows (Armonk, NY, USA). The Shapiro–Wilk normality test was performed to confirm the normal distribution of the indicators. The data were expressed as the mean plus standard error of the mean (M S.E.M.). Student’s t-test was performed to compare the mean values of SCK, SCA, and H, which were normally distributed variables. A probability of 0.05 was deemed significant (*p* < 0.05). Pearson’s correlation coefficient was determined to define the linear relationship between the variables under consideration. The statistical relationship between in-line milk fat-to-protein ratio and blood biochemical markers was determined using a linear regression equation. The relationship was considered statistically significant (*p* = 0.05) if the probability was less than 0.05. The cows were classified according to their milk fat-to-protein ratios: F/P < 1.2 (risk of SCA), F/P = 1.2 (healthy), and F/P > 1.5 (risk of SCK) [23].

## 3. Results

### Descriptive Statistics

In this study, we found significant differences between cows with SCK, cows with SCA, and healthy cows. Cows with SCK had higher in-line F/P ratios and NEFA concentrations, whereas cows with SCA had lower in-line F/P ratios as well as higher AST and GGT activities. The descriptive statistics are presented in Table 1. 

We found a higher level of milk fat-to-protein ratio in cows with SCK compared with SCA and H groups. The average in-line F/P ratio of SCK cows was significantly higher than that of SCA and H cows (*p* < 0.01). The average in-line F/P ratio of SCK cows was 1.66 (±0.29), that of SCA cows was 0.93 (±0.1), and that of healthy cows was 1.22. Cows with SCK had a 36% higher milk fat-to-protein ratio, while cows with SCA showed a decrease of 23.77% (Figure 3).

According to our results, we found a significantly higher AST activity of 26.66% in cows with SCA compared with the healthy group (*p* < 0.05). In SCA group, the average AST activity was 143.92 (±67.63) U/L, while in the healthy group, the average AST activity was 102.66 (±29.17) U/L (Figure 4).

We obtained a significantly higher GGT activity of 12.72% in cows with SCA compared to cows of the healthy group (*p* < 0.05). The average GGT activity in the SCA group was 39.07 (±14.68) U/L, while in the healthy group, the average GGT activity was 34.10 (±19.65) (Figure 5).

In cows with SCK, a significantly higher NEFA concentration of 40.38% was detected when compared to healthy cows (*p* < 0.001). The average NEFA concentration in the SCK group was 0.52 (±0.322) mmol/L, while in the healthy group, it was 0.31 (±0.25) mmol/L (Figure 6).

In–line milk F/P ratio had a strongly negative association with DIM (r = −0.363, *p* < 0.01), and a strongly positive correlation with blood NEFA concentration (0.499, *p* < 0.01) (Figure 7) and with blood glucose concentration (r = 0.287, *p* < 0.01) (Table 2).

Correlation between blood biochemical parameters. We found a positive correlation between blood BHBA and NEFA concentrations (r = 0.321, *p* < 0.01) and a negative association between BHBA and glucose concentrations (r = −0.330, *p* < 0.01). AST was strongly positively linked with GGT (r = 0.623, *p* < 0.01).

## 4. Discussion

Modern dairy farming frequently entails high milk production, which causes health problems in cows [24]. In this study, changes were compared in the in-line registered milk fat-to-protein ratio using a BROLIS HerdLine analyzer for the assessment of metabolic status in dairy cows. Based on our results, we found that cows with SCK had an in-line registered F/P ratio that was 36% higher and an NEFA concentration that was 40.38% higher than that of healthy cows. The results of the current study are in agreement with previous studies conducted by other researchers. According to the literature, the milk fat-to-protein ratio is commonly used to detect energy deficits [25] or subclinical ketosis [11]. A concomitant increase in milk fat-to-protein ratio correlates with NEB and adipose tissue mobilization. Milk F/P ratios of greater than 1.35 to 1.50 can be used to detect cows suffering from an energy shortfall [26]. Blood NEFA values are considerably higher in early lactation cows compared to mid, full, and late lactation cows, thus providing the best measure of negative energy balance (NEB) and lipomobilization during lactation [4,27]. Blood and milk serum concentrations of BHBA, further biomarkers of energy metabolism, are also considerably greater in early lactation cows than in the other groups of lactation cows, indicating strong fat reserve mobilization [6]. However, in this study, the mean BHBA was lower overall and decreased less across DIM when compared to previous research [28]. The NEFA concentrations, on the other hand, corresponded with the findings of Mäntysaari et al. [29] in the first three weeks of lactation in cows. We acknowledge that rumen fermentation involves a delicate balance of volatile fatty acids, including acetic acid and propionate, which play vital roles in the overall metabolism of dairy cows. Accordingly, we recommend future studies to investigate the relationship between milk F/P ratio, acetic acid, and propionate acid during rumen fermentation. Based on our findings and those of the literature, we may infer that in-line milk F/P ratio can be used to identify cows at a high risk of subclinical ketosis.

In our study, we found a lower F/P ratio by 23.77% in cows with SCA compared to healthy cows. Zschiesche et al. [20] found significant associations between milk F/P ratio and ruminal pH. Milk F/P ratio has been identified as a promising indicator to assess the SCA situation on a farm [30,31,32]. In fact, milk F/P ratio is frequently determined in dairy farming practice, and several research studies have confirmed the notion of F/P ratio as a good SARA indicator (for example, eight Holstein Friesian cows were reported to yield 25 kg of milk under trial conditions [33]). On the other hand, other studies did not establish a sufficiently robust link between F/P ratio and ruminal pH. For example, the limitations of F/P ratio as an SARA indicator were explored in a study involving 24 transition Holstein Friesian cows on a real farm [34] and another study involving Danish Holstein cows with 250 DIM in a trial setting [35]. Changes in F/P ratio were dependent on both a decrease in milk fat and an increase in milk protein content. A shift in volatile fatty acids (VFAs) in the rumen, with increased propionate and decreased acetate, has long been recognized as the cause of milk fat depression as a result of too many highly fermentable carbohydrates and insufficient structural effectiveness of diet [20]. Sutton [36] proposed that fluctuations in the molar proportions of VFAs in the rumen could explain up to 80% of the variation in milk fat. Furthermore, a decrease in milk fat synthesis due to specific products of ruminal fat biohydrogenation is now thought to provide a relatively solid explanation [37]. While the physiological principle underlying the relationship between low ruminal pH and low milk fat is well established, the same cannot be said for milk protein. While the decrease in protozoa commonly seen in SARA diets will increase the efficiency of bacterial growth via reduced predation, a low pH is associated with less efficient bacterial growth in general, which will have the opposite effect [30]. While the physiological basis is unclear, Plaizier et al. [14] also described an increased milk protein content in experimentally induced SARA, and Mensching et al. [30] confirmed the presence of a negative association between FPR and ruminal pH. Reduced F/P ratio, on the other hand, could be suggestive of (subacute) rumen acidosis because lower rumen volatile FA production (acetate and butyrate) serves as a precursor for mammary FA synthesis [38]. Individual trials, however, have demonstrated evidence for a connection between the two [39,40], and a threshold for SARA indication has also been established [32]. Mensching et al. [30] reported milk F/P ratio as an indicator for rumen pH parameters in a recent exploratory meta-analysis, but did not include it in the final prediction model. On the other hand, the limits of F/P ratio in SARA prediction have been consistently highlighted [31]. Based on our results and those reported in the literature, we can conclude that in-line F/P ratio can be used for the identification of cows with a higher risk of subclinical acidosis.

In this study, we found that the in-line F/P ratio had a strongly positive correlation with blood NEFA concentration (r = 0.499, *p* < 0.01). It is well recognized that the energy balance of dairy cows influences their milk composition, particularly during early lactation [6]. When experiencing NEB, cows mobilize their adipose tissue, which raises blood NEFA levels [41,42]. During the lactation period, an increase in NEFA supply for milk fat synthesis induces a rise in milk fat content and milk fat-to-protein ratio [43]. Milk fat can also be made from volatile fatty acids and, among other things, acetic acid generated in the rumen of cows [9]. A shortage of the main precursor, acetic acid, in the rumen causes low milk fat levels [1,24]. In previous studies, lactation performance and milk fat synthesis were improved by supplementing cows with branched-chain volatile fatty acids, which improved ruminal fermentation, nutritional digestibility, and the mRNA expression of genes involved in milk fat synthesis. Body fat is mobilized during NEB in early lactation, resulting in higher NEFA levels [44,45]. Cows must meet this increased energy demand by mobilizing fat from adipose tissue to supply nonesterified fatty acids (NEFA) as an energy fuel [42]. Excess fat mobilization raises the levels of NEFA in the blood. The majority of NEFA is utilized by end tissues and metabolized by hepatocytes via oxidation to acetyl-coenzyme A (Acetyl-CoA). Acetyl-CoA can alternatively be redirected to de novo cholesterol production or metabolized into ketones. Circulating ketone bodies can be used as a fuel source by the heart, brain, liver, and mammary tissue to some extent [42], but excessive ketogenesis and insufficient tissue absorption can result in increased circulating ketone bodies and, in rare cases, hyperketonemia [46]. BHBA is the most common circulating ketone body in ruminants [13], and its blood stability makes it a gold standard for diagnosing subclinical ketosis (SCK) [47]. Nevertheless, NEFA can be utilized to detect negative energy balance [48,49]. In a previous study, after confirming the presence of NEB at the start of lactation, an increase in the content of fat in milk, and a decrease in the protein content, the F/P ratio was proposed as a potential biomarker of insufficient dietary energy supply [26]. In addition to the level of NEFA [50] and diet, the fat content of milk might reflect ruminal health (for example, being affected by subclinical acidosis), which could have significant impacts during this period [14]. Toni et al. found that the F/P ratio is a useful indicator of lipomobilization and NEB in postpartum cows [51]. Considering the results of these studies and our results, we can state that the milk fat-to-protein ratio can serve as an indicator for assessing the metabolic condition of cows.

We found that blood AST activity was strongly positively linked with GGT activity (r = 0.623, *p* < 0.01). Other authors [4,9,52] have shown that milk enzyme activities can be good predictors of lipid mobilization and ketogenesis in cows during lactation to facilitate the early diagnosis of subclinical illnesses. This study’s high correlation coefficient agrees with the conclusion of Liu et al. [9], and the high significance can be attributed to the large number of samples evaluated in this experiment. Some milk enzyme activities were strongly positively correlated with negative energy balance (NEB) and lipomobilization biomarkers (NEFA and BHBA), as well as blood biomarkers of liver excretory capacity (TBil), and were negatively correlated with liver functional state parameters (glucose, TChol, and TG) [43]. Dynamic changes in metabolite values in blood and milk during lactation are identical, which confirms the possibility of using metabolic parameters from milk for the purpose of assessing the metabolic status of cows [43,53]. Correlation, regression, and covariance analyses of blood and milk metabolic parameters confirm that milk metabolic parameters can be used as indicators to evaluate the metabolic status of cows. The metabolic parameters in milk are significant indicators of the metabolic stress of cows because they correlate with the parameters of lipolysis and ketogenesis in the blood of cows. It is necessary to know the lactation period to correctly interpret the mutual relationships between these metabolic parameters. Milk samples are obtained non-invasively, which additionally makes it suitable for evaluating the metabolic status of cows during routine dairy farming practice when milk comes from a healthy udder [43]. The liver, being a primary organ involved in ruminant metabolism, exhibits sensitivity to alterations in nutrition. Serum AST and GGT are commonly employed as indicators of hepatic injury caused by metabolic disorders or external stresses. The levels of AST and GGT in the bloodstream are elevated in cases of liver injury, leading to the release of these intracellular enzymes into the serum [54]. Furthermore, it has been established that metabolic stress in cows during the early lactation period has an impact on liver health. This is attributed to the negative energy balance, which triggers heightened lipolysis, resulting in excessive lipid accumulation and the development of liver lesions. Consequently, this process leads to an elevation of liver enzymes [55]. In our study, it was observed that cows affected by SCA had elevated levels of AST and GGT activities. The results obtained in our study are in agreement with the results obtained by Morar et al., which also showed that the activities of AST and glutamate dehydrogenase (GLDH) were significantly higher in cows with spontaneous subacute ruminal acidosis (SARA) (*p* < 0.05) than in healthy cows [56]. Elevated levels of AST in the bloodstream are regarded as a highly responsive indicator for the detection of hepatocellular injury, even in cases where the injury is not readily apparent [57]. According to the available scientific data, dairy cows demonstrate the highest level of AST activity during the initial phase of lactation. Nevertheless, as the duration of lactation progresses, a decrease in the activity of this specific enzyme is seen [58].

Based on the results of our study and other studies, we can conclude that the milk fat-to-protein ratio corresponds to lipolysis and ketogenesis indicators in cow blood and is an important predictor of metabolic stress in cows. According to Negussie et al., the determination of the milk fat-to-protein ratio can be achieved through the regular practice of milk recording [59]. The results of our study highlight the significance of milk as a potentially advantageous medium for the detection of subclinical metabolic disorders, including ketosis and acidosis. The convenience of milk sampling and the ability to conduct whole herd-level testing during routine recording contribute to its potential as a diagnostic matrix. The assessment of fat-to-protein ratio is non-invasive and can be conducted during each milking session without inducing further distress to the bovines. To facilitate the comparison of results in future studies, we recommend measuring the in-line milk fat-to-protein ratio along with BHBA, triglycerides, cortisol, high-density lipoproteins, and very-low-density lipoproteins [60]. A weakness of our study is that the SCA group had a small number of animals because, on this farm during the research period, SCA was not a very frequently seen health problem in the cows. Because of this, in future studies, we recommend investigating changes in in-line registered milk fat-to-protein ratio for the assessment of metabolic status in dairy cows using a large number of animals. 

## 5. Conclusions

It can be concluded that changes in the in-line milk fat-to-protein ratio can be used for the identification of metabolic status in dairy cows. In-line F/P ratio can be used for the identification of cows with a higher risk of NEB because in-line F/P ratio has a strongly positive relationship with blood NEFA concentration. Also, the in-line F/P ratio can be used for the identification of cows with a higher risk of subclinical ketosis and subclinical acidosis. Cows with a higher risk of SCK had a higher milk F/P ratio of 36%, whereas cows with a higher risk of SCA had a lower ratio of 23.77% than cows without a risk of these diseases. Based on these results, we can state that the comparison of blood and milk metabolic data reveals that the milk fat-to-protein ratio can be used to assess cow metabolic condition. Milk characteristics are a key predictor of metabolic stress in cows because they correspond with indicators of lipolysis and ketogenesis in cow blood. Milk samples are taken non-invasively, making them suitable for assessing metabolic status in routine practice.

We believe that this could be a beneficial complement to dairy farm herd health programs, and monitoring individual cow energy status enables farmers to determine which cows are at risk of metabolic stress.

## Figures and Tables

**Figure 1 animals-13-03293-f001:**
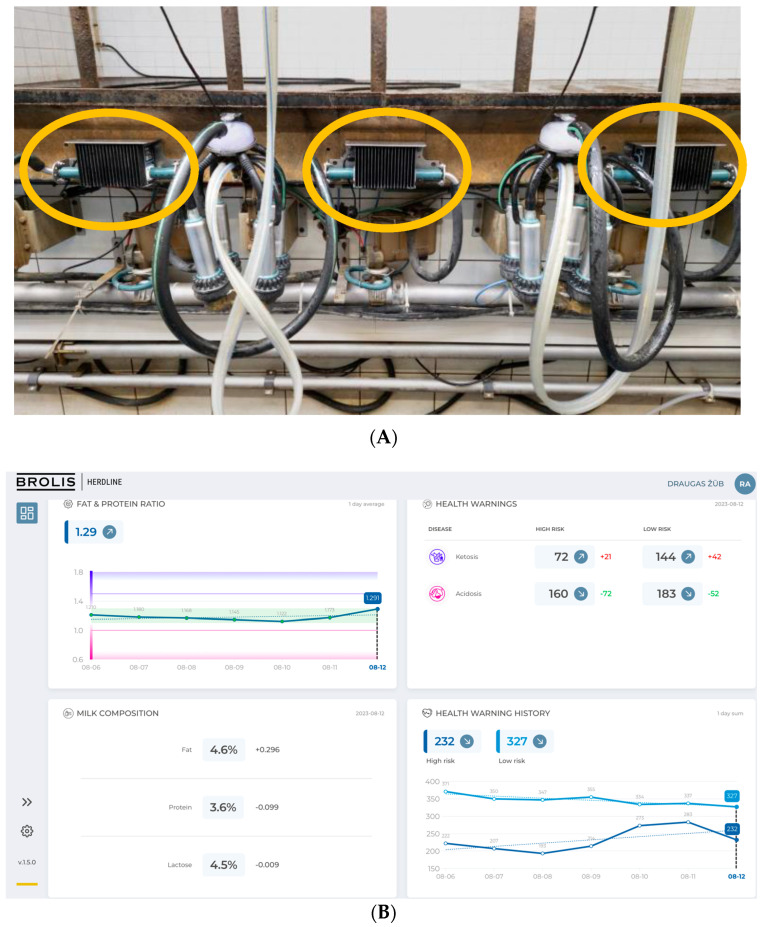
(**A**) The in-line milk analyzer BROLIS HerdLine, and (**B**) registration and analysis of data (Brolis Sensor Technology, Vilnius, Lithuania).

**Figure 2 animals-13-03293-f002:**
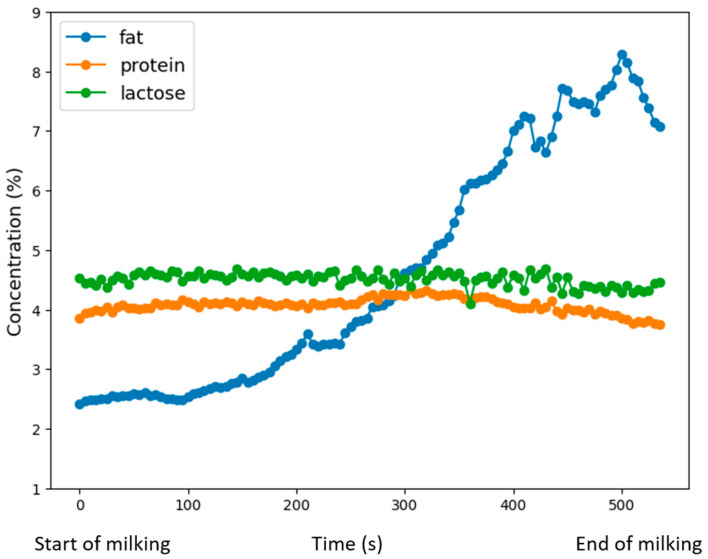
Dynamic measurement of an in-line milk analyzer BROLIS HerdLine (Brolis Sensor Technology, Vilnius, Lithuania).

**Figure 3 animals-13-03293-f003:**
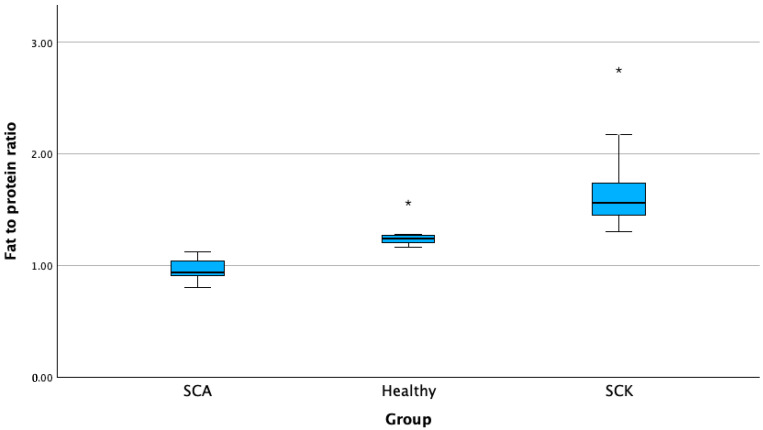
In-line milk fat-to-protein ratio in cows with SCK, cows with SCA, and healthy cows. SCA—subclinical acidosis; Healthy—healthy cows; SCK—subclinical ketosis. * *p* < 0.01.

**Figure 4 animals-13-03293-f004:**
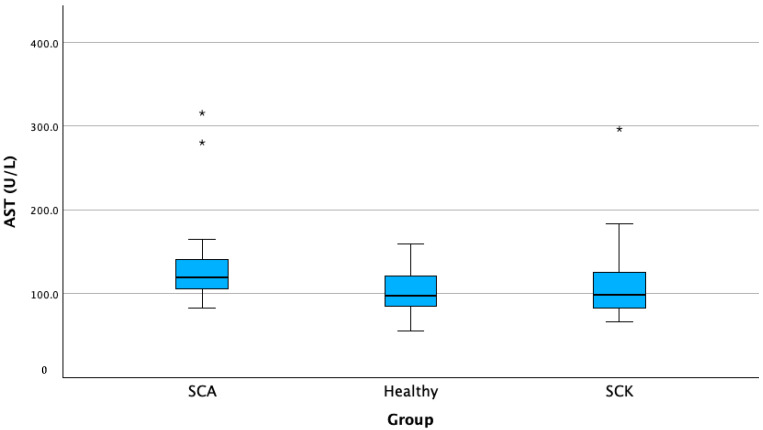
AST activity in cows with SCK, cows with SCA, and healthy cows. SCA—subclinical acidosis; Healthy—healthy cows; SCK—subclinical ketosis; AST—aspartate transaminase. * *p* < 0.01, ** *p* < 0.05.

**Figure 5 animals-13-03293-f005:**
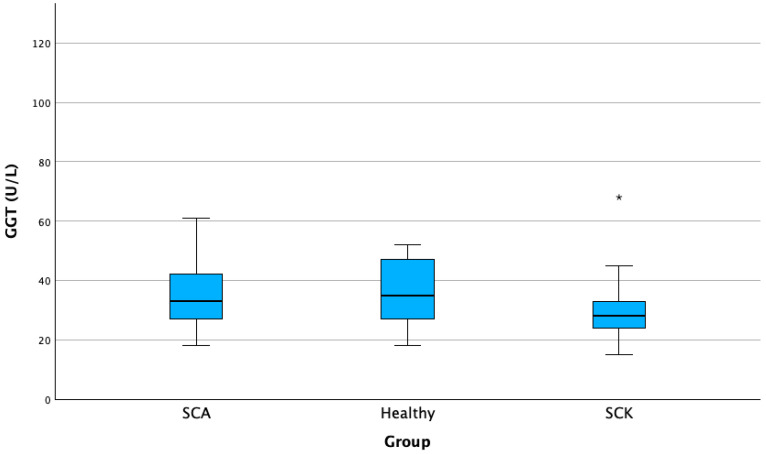
AST activities in cows with SCK, cows with SCA, and healthy cows. SCA—subclinical acidosis; Healthy—healthy cows; SCK—subclinical ketosis; GGT—gamma-glutamyltransferase. * *p* < 0.01.

**Figure 6 animals-13-03293-f006:**
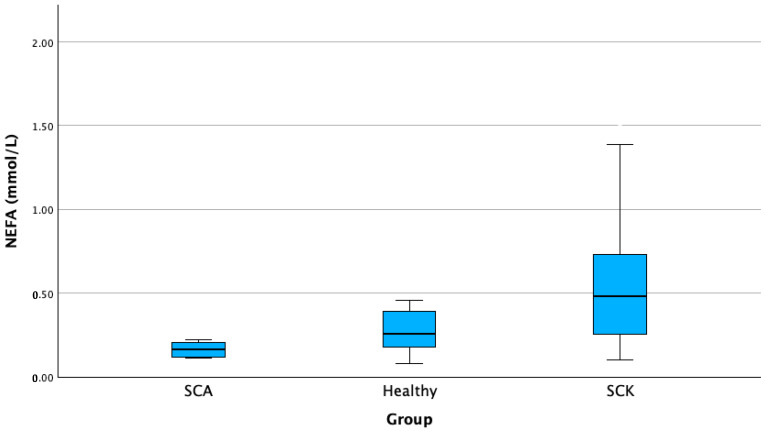
NEFA concentration in cows with SCK, cows with SCA, and healthy cows. SCA—subclinical acidosis; Healthy—healthy cows; SCK—subclinical ketosis; NEFA—nonesterified fatty acids.

**Figure 7 animals-13-03293-f007:**
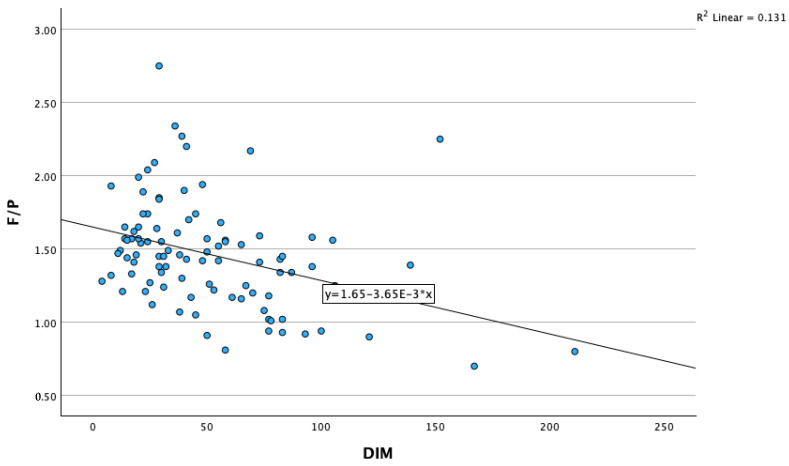
Correlation of in-line milk F/P ratio with NEFA concentration. F/P—in-line milk fat-to-protein ratio; NEFA—nonesterified fatty acids.

**Table 1 animals-13-03293-t001:** Descriptive statistics of milk fat-to-protein ratio and biochemical markers.

		N	Mean	Standard Deviation	Std. Error	95% Confidence Interval for Mean		Minimum	Maximum
						Lower Bound	Upper Bound		
Milk fat-to-protein ratio	SCK ^A^	62	1.66 ^B,C^	0.29	0.00	1.58	1.73	1.33	2.75
	H ^B^	20	1.22	0.07	0.01	1.19	1.26	1.07	1.34
	SCA^C^	14	0.93	0.10	0.02	0.86	0.99	0.70	1.08
	Total	96	1.46	0.36	0.03	1.38	1.53	0.70	2.75
Capillary glucose (mmol/L)	SCK	61	2.83	0.32	0.04	2.75	2.92	2.1	3.6
	H	20	2.82	0.29	0.06	2.68	2.96	2.4	3.5
	SCA	14	2.58	0.46	0.12	2.31	2.85	1.1	3.0
	Total	95	2.79	0.35	0.03	2.72	2.87	1.1	3.6
Capillary BHBA (mmol/L)	SCK	62	0.42	0.35	0.0451	0.336	0.516	0.1	1.8
	H	20	0.32	0.12	0.0289	0.265	0.385	0.1	0.5
	SCA	14	0.45	0.44	0.1189	0.200	0.714	0.1	1.9
	Total	96	0.40	0.33	0.0344	0.341	0.478	0.1	1.9
DIM	SCK	62	42.79	30.03	3.81	35.16	50.42	5	29
	H	20	45.85	27.32	6.11	33.06	58.64	7	30
	SCA	14	94.14	45.45	12.14	67.90	120.38	9	28
	Total	96	50.92	36.54	3.72	43.51	58.32	7	29
Serum AST (U/L)	SCK ^A^	62	109.72	40.58	5.15	99.41	120.03	66.0	296.6
	H ^B^	20	102.66	29.17	6.52	89.01	116.31	55.2	159.6
	SCA ^C^	14	143.92 ^B^	67.63	18.07	104.87	182.98	95.0	315.7
	Total	96	113.24	44.99	4.59	104.12	122.35	55.2	315.7
Serum GGT (U/L)	SCK ^A^	62	30.65	9.62	1.22	28.20	33.09	15	68
	H ^B^	20	34.10	19.65	4.39	24.90	43.30	16	105
	SCA ^C^	14	39.07 ^B^	14.68	3.92	30.59	47.55	24	78
	Total	96	32.59	13.24	1.35	29.91	35.28	15	105
Serum NEFA (mmol/L)	SCK ^A^	62	0.527 ^B^	0.32	0.04	0.44	0.60	0.10	1.52
	H ^B^	20	0.316	0.25	0.05	0.19	0.43	0.08	1.12
	SCA ^C^	14	0.183	0.08	0.02	0.13	0.23	0.11	0.41
	Total	96	0.433	0.31	0.03	0.36	0.49	0.08	1.52
Serum albumin (g/L)	SCK	62	35.98	2.23	0.28	35.41	36.55	29.2	39.9
	H	20	36.16	2.09	0.46	35.18	37.14	31.1	39.4
	SCA	14	36.87	1.97	0.52	35.73	38.02	32.9	40.4
	Total	96	36.15	2.17	0.22	35.71	36.59	29.2	40.4
Lactation number	SCK	62	2.27	1.681	0.213	1.85	2.70	2	4
	H	20	2.00	1.124	0.251	1.47	2.53	2	5
	SCA	14	1.93	0.730	0.195	1.51	2.35	2	3
	Total	96	2.17	1.470	0.150	1.87	2.46	2	4

The letters ^A^, ^B^, and ^C^ show statistically significant differences in the mean values across groups; SCA—subclinical acidosis; H—healthy cows; SCK—subclinical ketosis; BHBA—β-hydroxybutyrate; AST—aspartate transaminase; GGT—gamma-glutamyltransferase; NEFA—nonesterified fatty acids; DIM—days in milk.

**Table 2 animals-13-03293-t002:** Correlation between in-line milk fat-to-protein ratio and blood biochemical parameters.

Correlations										
		Fat-to-Protein Ratio	Serum GGT	Serum AST	Lactation Number	DIM	Capillary Glucose	Capillary BHBA	Serum NEFA	Serum Albumin
Fat-to-protein ratio	Pearson‘s correlation	1	–0.161	–0.052	0.171	–0.363 **	0.287 **	0.195	0.499 **	–0.282 **
	sig. (two-tailed)		0.118	0.612	0.096	<0.001	0.005	0.057	<0.001	0.005
	N	96	96	96	96	96	95	96	96	96
Serum GGT	Pearson’s correlation	–0.161	1	0.623 **	0.179	0.273 **	0.059	–0.109	0.072	–0.107
	sig. (two-tailed)	0.118		<0.001	0.075	0.006	0.571	0.289	0.477	0.288
	N	96	100	100	100	100	96	97	100	100
Serum AST	Pearson’s correlation	–0.052	0.623 **	1	0.115	0.140	–0.028	0.105	0.258 **	–0.053
	sig. (two-tailed)	0.612	<0.001		0.255	0.166	0.783	0.308	0.009	0.603
	N	96	100	100	100	100	96	97	100	100
Lactation number	Pearson’s correlation	0.171	0.179	0.115	1	0.035	–0.210 *	0.329 **	0.135	–0.077
	sig. (two-tailed)	0.096	0.075	0.255		0.731	0.040	0.001	0.179	0.445
	N	96	100	100	100	100	96	97	100	100
DIM	Pearson’s correlation	–0.363 **	0.273 **	0.140	0.035	1	–0.052	–0.155	–0.460 **	0.084
	sig. (two-tailed)	<0.001	0.006	0.166	0.731		0.618	0.129	<0.001	0.407
	N	96	100	100	100	100	96	97	100	100
Capillary glucose	Pearson’s correlation	0.287 **	0.059	–0.028	–0.210 *	–0.052	1	–0.330 **	0.156	–0.003
	sig. (two-tailed)	0.005	0.571	0.783	0.040	0.618		0.001	0.128	0.979
	N	95	96	96	96	96	96	96	96	96
Capillary BHBA	Pearson’s correlation	0.195	–0.109	0.105	0.329 **	–0.155	–0.330 **	1	0.321 **	–0.123
	sig. (two-tailed)	0.057	0.289	0.308	0.001	0.129	0.001		0.001	0.229
	N	96	97	97	97	97	96	97	97	97
Serum NEFA	Pearson’s correlation	0.499 **	0.072	0.258 **	0.135	–0.460 **	0.156	0.321 **	1	–0.073
	sig. (two-tailed)	<0.001	0.477	0.009	0.179	<0.001	0.128	0.001		0.468
	N	96	100	100	100	100	96	97	100	100
Serum albumin	Pearson’s correlation	–0.282	–0.107	–0.053	–0.077	0.084	–0.003	–0.123	–0.073	1
	sig. (two-tailed)	0.005	0.288	0.603	0.445	0.407	0.979	0.229	0.468	
	N	96	100	100	100	100	96	97	100	100

* Correlation is significant at the 0.05 level (2-tailed); ** Correlation is significant at the 0.01 level (2-tailed); BHBA—β-hydroxybutyrate; AST—aspartate transaminase; GGT—gamma-glutamyltransferase; NEFA—nonesterified fatty acids; DIM—days in milk.

## Data Availability

The data provided in this study can be found in the publication.

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
