# Peer review of "In-Line Registered Milk Fat-to-Protein Ratio for the Assessment of Metabolic Status in Dairy Cows"

_animals, 2023, doi:10.3390/ani13203293_

Round 1

Reviewer 1 Report

The manuscript number 2584422 received from ANIMALS, with title: “In-line Registered Milk Fat/Protein Ratio For The Assessment Of Metabolic Status In Dairy Cows has been revised.

The paper has some points of interest; however, some critical issues remain that need to be clarified and/or implemented:

- the English language is rather correct, the only suggestion it is that in a scientific paper it is better avoid the use of personal form as (row 252): “we compared changes….. “ “changes have been compared ….”. please : check and modify also the rest in the paper;

- authors assert that the number of observations it is a vast number (row 323). it seems not appropriate for this kind of investigation, and the numerosity of groups is imbalanced (62 vs 20 vs 14);

- in figures numbers and symbols are reported: describe in footnote the meaning or remove.

- description of the diet it is a bit messy: there is not the % for sugar beet pulp silage; what is grass hay wheat straw? In the description of the gross composition sometime appear “% symbol” sometime “as percentage”, please standardize.  Is there any integration with mineral and vitamin?

- authors indicate that the average energy milk yield was 12,500 kg per lactation but is a std lactation of 305 d, or complete lactation? In the latter case please indicate the length. The milk yield seems a bit high, particularly with a so high content of fat, content that appears not justified by the low content of fiber and high content in non fiber carbohydrates (please correct at row 107) in the diet

- also description of time (in particular distance from TMR deliver), site and technics of bleeding are a messy: please use this order followed by plasma preparation and at the end the description of the analysis

Row               comment

120-121        remove the sentence “It’s similar to having a small laboratory on your dairy farm”, it is not proper of a scientific paper.

151                indigestion),

160                “During the whole investigation period, no cows were sick or healthy.” What does it mean?

166                “ The data were expressed as the mean standard error of the mean” it is not clear, I have never met before

Results          table 1: concentration of glucose it is very low, usually glucose is around 3,5 mmol/L if collected from jugular vein before feed delivery. Are blood from tail adequate to evaluate metabolic condition?

                       the correlation between parameters determined with stik (figure 7 A & B) in my opinion does not have sense

discussion    the main lack of this section it is that authors never consider the fact that besides the low acetic acid yield in the rumen (also BHB is used for fattyacid synthesis) milk fat is depressed also by the fact that high propionate yield in the rumen leads to a high glucose yield in the liver that stimulates high insulin release that blocks lipolysis in adipose tissue (i.e. less NEFA) and  stimulate peripheral tissues to use nutrient decreasing their availability for mammary gland.

                       Correlation between NEFA and F/P is quite strong, anyway looking to the distribution of the data (nearly half of the data between 0,2 and 0,5 mmol/L are in a vertical shape) I suggest authors not to give so large importance and discussion

                       Author do not discuss why in in cows of SCA group AST and GGT concentration was significantly higher (Table 1)

268                “The milk F/P is an indication of tissue that may be acquired for free from routine milk recording” This statement is not clear

270                R/B : what is?

282                LDA: what is?

322-324        the statement it is not comprehensible

327                sure that glucose is a parameter to measure liver functional state?

328                “Dynamic changes in metabolite values in blood and milk during lactation are identical” more than one citation are needed to support the statement

some wording should be improved

Author Response

Dear Reviewer, 

Authors are very thankful for the comments, which help us to improve the manuscript. All changes proposed have been included in the manuscript and highlighted in yellow and track changes.  

Best Regards, 

Prof. Ramunas Antanaitis 

Question  

Answers  

The manuscript number 2584422 received from ANIMALS, with title: “In-line Registered Milk Fat/Protein Ratio For The Assessment Of Metabolic Status In Dairy Cows has been revised.

The paper has some points of interest; however, some critical issues remain that need to be clarified and/or implemented:

Thank you for your positive comment and recommendations for the manuscript. We corrected the manuscript accordingly.

- the English language is rather correct, the only suggestion it is that in a scientific paper it is better avoid the use of personal form as (row 252): “we compared changes….. “ “changes have been compared ….”. please : check and modify also the rest in the paper;

The personal form (line 268) “we compared changes…” was change into “changes have been compared”.

Line 192, 193. Sentence “Throughout the study period, we classified cows' clinical state as healthy or unwell.” was corrected into “Throughout the study period, cows' clinical states were classified as healthy or unwell.”

Line 209 – 211. Sentence “In cows with SCK, we determined higher in-line F/P and NEFA concentrations, in cows with SCA, we identified lower in-line F/P values as well as higher AST and GGT activities.” was corrected into “Cows with SCK had higher in-line F/P and NEFA concentrations, whereas cows with SCA had lower in-line F/P values as well as higher AST and GGT activities.”

Line 254. Sentence “In cows with SCK, we detected a significant 40.38% higher NEFA concentration (p<0.001).” was corrected into “In cows with SCK, a significant 40.38% higher NEFA concentration has been detected.”

- authors assert that the number of observations it is a vast number (row 323). it seems not appropriate for this kind of investigation, and the numerosity of groups is imbalanced (62 vs 20 vs 14);

We do not find such information in row 323. However, we corrected the information following:

We added information – „Clinical examinations were performed on all 320 cows. According to the results of the clinical examination and blood BHBA from 320 cows for this study, 96 cows were selected: in the SCK group, 62 were selected for SCA, 14, and in the healthy group, 34 cows. The same cows were examined at each visit. 14 cows that at the start of the study were assigned to the healthy group and during the experiment became sick were removed from this study. In this case, the final number in the healthy group was 20

Also, small number of cows in SCA was because  that during exeperiment in this farm SCA was not very often problem. In methods section we added – “Clinical examinations were performed on all 320 cows. According to the results of the clinical examination and blood BHBA from 320 cows for this study, 96 cows were selected: in the SCK group, 62 were selected for SCA, 14, and in the healthy group, 34 cows. The same cows were examined at each visit. 14 cows that at the start of the study were assigned to the healthy group and during the experiment became sick were removed from this study. In this case, the final number in the healthy group was 20

According to this, in discussion section we added – “Also, as a weakness of our study, we can declare that the SCA group had a small number of animals because, on this farm during the research period, SCA was not very often cows health problem. Because of this, in future studies, we recommend investigating changes in the in-line registered milk fat to protein ratio for the assessment of metabolic status in dairy cows with the large number of animals”

- in figures numbers and symbols are reported: describe in footnote the meaning or remove.

Corrected, all figures, numbers, and symbols are removed.

- - description of the diet it is a bit messy: there is not the % for sugar beet pulp silage; what is grass hay wheat straw? In the description of the gross composition sometime appear “% symbol” sometime “as percentage”, please standardize.  Is there any integration with mineral and vitamin?

We corrected to – “ Cows were fed at 06:00 and 18:00 every day, with a typical total mixed ration for high-producing, multiparous cows consisting primarily of 25% corn silage, 5% alfaalfa grass hay, 20% grass silage, 15% sugar beet pulp silage, 30% grain concentrate mash, and 5% mineral mixture”

In all paper “%” and “percentage” was standardized into “%”.

….

- authors indicate that the average energy milk yield was 12,500 kg per lactation but is a std lactation of 305 d, or complete lactation? In the latter case please indicate the length. The milk yield seems a bit high, particularly with a so high content of fat, content that appears not justified by the low content of fiber and high content in non fiber carbohydrates (please correct at row 107) in the diet

Line 137. Sentence “The average energy corrected milk yield (4.2% fat, 3.6% protein) per cow per lactation was 12,500 kg.” was corrected into “The average energy corrected milk yield (4.2% fat, 3.6% protein) per cow per lactation (305 days) was 12,500 kg.”

- also description of time (in particular distance from TMR deliver), site and technics of bleeding are a messy: please use this order followed by plasma preparation and at the end the description of the analysis

Line 164 - 169. Blood collection techniques, preparation, and analysis were improved.

120-121        remove the sentence “It’s similar to having a small laboratory on your dairy farm”, it is not proper of a scientific paper.

The sentence “It’s similar to having a small laboratory on your dairy farm” was removed.

151                indigestion),

Line 184 and 189. This has been fixed.

160                “During the whole investigation period, no cows were sick or healthy.” What does it mean?

This sentence was corrected into: “During the whole investigation period, all 20 cows were clinically healthy.”

166                “ The data were expressed as the mean standard error of the mean” it is not clear, I have never met before

It is not clear what you mean. L 166 presents information about group creation. If you are talking about the statistical analysis part, we usually present results by means and standard deviation. 

Results          table 1: concentration of glucose it is very low, usually glucose is around 3,5 mmol/L if collected from jugular vein before feed delivery. Are blood from tail adequate to evaluate metabolic condition?

During the farm visit, blood samples from 320 cows were gathered and glucose and BHB concentrations measured. Plasma concentrations of BHB and glucose levels were assessed using a capillary bloodsample obtained from the ear during a clinical examination.

According our results (Table 1), level of glucose in healthy cows was from 2.4 – 3.5 mmol/L, in SCA group – 2.1 – 3.6 mmol/L, in SCK group – 1.1 – 3.0 mmol/L.

According to the literature glucose level - 2.6–3.8 mmol/L -   Cozzi, G., Ravarotto, L., Gottardo, F., Stefani, A. L., Contiero, B., Moro, L., ... & Dalvit, P. (2011). Reference values for blood parameters in Holstein dairy cows: Effects of parity, stage of lactation, and season of production. Journal of dairy science94(8), 3895-3901

       th  the correlation between parameters determined with stik (figure 7 A & B) in my opinion does not have sense

We deleted Fig 6 A and C and Fig 7 A, B and C.

discussion    the main lack of this section it is that authors never consider the fact that besides the low acetic acid yield in the rumen (also BHB is used for fattyacid synthesis) milk fat is depressed also by the fact that high propionate yield in the rumen leads to a high glucose yield in the liver that stimulates high insulin release that blocks lipolysis in adipose tissue (i.e. less NEFA) and  stimulate peripheral tissues to use nutrient decreasing their availability for mammary gland.

Inn discussion section we added – “Zschiesche et al.[20], found significant associations between milk F/P and ruminal pH. F/P has been identified as a promising indicator in the effort to assess  the SCA situation on a farm [30], [31], [32]. In fact, F/P is frequently regarded in dairy practice, and several research confirm the notion of F/P as a good SARA indicator (for example, 8 Holstein Friesian cows yielding 25 kg of milk under trial conditions [33]. Others, on the other hand, did not establish a sufficiently tight link between F/P and ruminal pH (for example, 24 transition Holstein Friesian cows on a real farm [34]Danish Holstein cows with 250 DIM in a trial setting [35], and the limitations of FPR as a SARA indicator are explored. Changes in F/P were dependent on both a decrease in milk fat and an increase in milk protein content. A shift of volatile fatty acids (VFAs) in the rumen with increased propionate and decreased acetate has long been recognized as the cause of milk fat depression as a result of too many highly fermentable carbohydrates and insufficient structural effectiveness in the diet [20]. Sutton [36] proposed that fluctuations in the molar proportions of VFAs in the rumen could explain up to 80% of the variation in milk fat. Furthermore, a decrease in milk fat synthesis due to specific products of ruminal fat biohydrogenation is now thought to be a relatively solid explanation [37]. While the physiological principle underlying the relationship between low ruminal pH and low milk fat is well established, the same cannot be said for milk protein. While the decrease in protozoa commonly seen in SARA diets will increase the efficiency of bacterial growth via reduced predation, a low pH is associated with less efficient bacterial growth in general, which will have the opposite effect [30]. While the physiological basis is unclear, Plaizier et al. [14], who also described an increased milk protein content in experimentally induced SARA, or Mensching et al. [30] confirmed the presence of a negative association between FPR and ruminal pH

Also, we added -

“We acknowledge that rumen fermentation involves a delicate balance of volatile fatty acids, including acetic acid and propionate, which play vital roles in the overall metabolism of dairy cows. Accordingly, we recommend future studies investigate the relationship between milk F/P, acetic acid, and propionate acid during rumen fermentation”

Correlation between NEFA and F/P is quite strong, anyway looking to the distribution of the data (nearly half of the data between 0,2 and 0,5 mmol/L are in a vertical shape) I suggest authors not to give so large importance and discussion

We deleted a few sentences from the discussion section that related to these results. 

 Author do not discuss why in in cows of SCA group AST and GGT concentration was significantly higher (Table 1)

A new paragraph has been added to the discussion area (Line 408 - 426).

268                “The milk F/P is an indication of tissue that may be acquired for free from routine milk recording” This statement is not clear

Line 331. The sentence “The milk F/P is an indication of tissue that may be acquired for free from routine milk recording” corrected into “The determination of the milk fat to protein ratio can be achieved through the regular practice of milk recording.”

270                R/B : what is?

“R/B” was corrected into “F/P” (Line 289).

282                LDA: what is?

“LDA” was corrected into “LAD” in all paper.

LAD means left abomasal displacement.

322-324        the statement it is not comprehensible

We deleted this sentence

327                sure that glucose is a parameter to measure liver functional state?

Glucose is not typically used as a direct parameter to measure the functional state of the liver. In dairy cows, glucose levels can be used as one of several parameters to assess the metabolic and health status of the animal, but it is not a direct measure of liver function in the same way it might be in humans. In dairy cow management, glucose levels are often monitored as part of a broader evaluation of the cow's metabolic health and energy balance.

However, while glucose levels can provide valuable information about the cow's overall health and metabolic state, they are not a direct measure of liver function like specific liver enzyme tests or imaging studies. For example, in advanced liver disease (such as ketosis or a negative energy balance), the liver may have difficulty in producing and regulating glucose, which can lead to abnormalities in blood glucose levels. In these cases, glucose levels can be used as part of the overall assessment of a cow with liver disease, but they are not a direct measure of liver function.

328                “Dynamic changes in metabolite values in blood and milk during lactation are identical” more than one citation are needed to support the statement

Line 399. The citation “Dynamic changes in metabolite values in blood and milk during lactation are identical” was supported for two more references.

Comments on the Quality of English Language some wording should be improved

Corrected by the MDPI service. Certificate added. 

Reviewer 2 Report

Dear Editor and Authors,

I send you my review about the paper “In-line Registered Milk Fat/Protein Ratio For The Assessment Of Metabolic Status In Dairy Cows”.

The scope of the paper, as reported in the aim was to ascertain alterations in the in-line registered milk fat to protein ratio as a potential indicator for evaluating the metabolic status of dairy cows.

The paper result sufficiently well written and well structured, however, in this form it show some little lacks that I reported below.

Although the paper is well written, a moderate English language revision should be necessary. Indeed, in my opinion, to facilitate the reader's understanding of the text it would be better to structure the sentences in the passive form.

Futhermore, in the title should replace "Fat/Protein" with "Fat to Protein"

Regarding the introduction, it result well written and adequately to the aim of the paper.

However, at line 55 the acronym “NEFA” should be explicit.

Moreover, in the introduction, and in all other parts of the text, should be replaced “fat/protein ratio” with “fat to protein ratio”.

In addition, at line 63, should be replaced the symbols “>” with “over to”.

The experimental design is well structured and adequate to the aim.

Nevertheless, to improve the reading by the readers, in this section, the number of cow followed should be reported.

Moreover, should be reported, also, the number of trials performed and the number of samples collected in each trials for each thesis.

The results is well presented and well discussed, also, in relation to the references reported.

However, each table with its captions should be placed in a single page.

Moreover, in the captions of the tables and figures should be avoid the use of acronyms like “F/P”.

Finally, the conclusions of the paper, are adequate to the results showed and they satisfy the aim of the research.

Nevertheless, the section of the conclusions should not be limited only to reporting a summary of the data already reported, but should also include some personal comments of the Authors.

In this regard, I would suggest that the authors report their opinion on the impact that the results of their paper could have.

Dear Editor and Authors,

In my opinion, although the paper is well written, a moderate English language revision should be necessary.

Indeed, to facilitate the reader's understanding of the text, it would be better to structure the sentences in the passive form.

Moreover, the Authors should change the citation method of two or more article.

For examples, the text “[1], [2]” should be change in “[1,2]”, please check the Authors guide line.

Author Response

Dear Reviewer, 

Authors are very thankful for the comments, which help us to improve the manuscript. All changes proposed have been included in the manuscript and highlighted in yellow and track changes.  

Best Regards, 

Prof. Ramunas Antanaitis 

Question  

Answers  

 The scope of the paper, as reported in the aim was to ascertain alterations in the in-line registered milk fat to protein ratio as a potential indicator for evaluating the metabolic status of dairy cows.

The paper result sufficiently well written and well structured, however, in this form it show some little lacks that I reported below.

Although the paper is well written, a moderate English language revision should be necessary. Indeed, in my opinion, to facilitate the reader's understanding of the text it would be better to structure the sentences in the passive form.

Thank you for your positive comment on the article.

Futhermore, in the title should replace "Fat/Protein" with "Fat to Protein"

“Fat/Protein” in the title was replaced into “Fat to Protein”.

Regarding the introduction, it result well written and adequately to the aim of the paper.

Thank you for your positive comment on the article.

However, at line 55 the acronym “NEFA” should be explicit.

Line 70 the acronym “NEFA” was explained into “nonesterified fatty acids”.

Moreover, in the introduction, and in all other parts of the text, should be replaced “fat/protein ratio” with “fat to protein ratio”.

In all paper “fat/protein ratio” was replaced into “fat to protein ratio”.

Line 28, 43, 45, 56, 79, 113, table 1, 212, 214, 219, 264, 284, 288, 289, 328, 380, 388

In addition, at line 63, should be replaced the symbols “>” with “over to”.

Line 82 symbols “>” was corrected into “over to”

The experimental design is well structured and adequate to the aim.

Thank you for your positive comment on the article.

Nevertheless, to improve the reading by the readers, in this section, the number of cow followed should be reported.

Moreover, should be reported, also, the number of trials performed and the number of samples collected in each trials for each thesis.

We added information – “From 1160 cows for clinical examination, 320 were selected (second and more lactation from the first 5 to30 days after calving)”

And – “Throughout the study period, the farm was visited two times per week on the same days (on Tuesdays and Thursdays) each week. According to the milk fat-to-protein ratio registered by in -line analyzer (Brolis Sensor Technology, Vilnius, Lithuania of 1160 cows, 320 (second and more lactation from the first 5 to 30 days after calving) were selected. Clinical examinations were performed on all 320 cows. According to the results of the clinical examination and blood BHB from 320 cows for this study, 96 cows were selected: in the SCK group, 62 were selected for SCA, 14, and in the healthy group, 20 cows. The same cows were examined at each visit. Cows that at the start of the study were assigned to the healthy group and during the experiment became sick were removed from this study”

The results is well presented and well discussed, also, in relation to the references reported.

Thank you for your positive comment on the manuscriopt.

However, each table with its captions should be placed in a single page.

Thank you for suggestion, it will be corrected during final proof.

Moreover, in the captions of the tables and figures should be avoid the use of acronyms like “F/P”.

Line 257 “F/P” was corrected into “fat to protein ratio”.

Table 2: acronym “F/P” was corrected into “Fat to protein ratio”.

Finally, the conclusions of the paper, are adequate to the results showed and they satisfy the aim of the research.Nevertheless, the section of the conclusions should not be limited only to reporting a summary of the data already reported, but should also include some personal comments of the Authors.

In conclusions section we added – “We think that comparing the metabolic data of blood and milk reveals that the milk fat-to-protein ratio can serve as an indicator for assessing the metabolic condition of cows. Milk characteristics play a pivotal role in predicting cow metabolic stress, as they correspond to indicators of lipolysis and ketogenesis in cows. Furthermore, the non-invasive methods of milk sample collection make it a suitable method for routine assessment of metabolic status.We believe that this could be a beneficial complement to dairy farm herd health programs, and monitoring individual cow energy status allows farmers to determine which cows are at risk of metabolic stress”

In this regard, I would suggest that the authors report their opinion on the impact that the results of their paper could have.

Added authors opinion in discussion part:

“This work highlights the significance of milk as a potentially advantageous medium for the detection of subclinical metabolic disorders, including ketosis and acidosis. The convenience of milk sampling and the ability to conduct whole herd testing during routine recording contribute to its potential as a diagnostic matrix. The assessment of fat to protein ratio is non-invasive and can be conducted during each milking session without inducing further distress to the bovine.“

Comments on the Quality of English Language

Dear Editor and Authors,

In my opinion, although the paper is well written, a moderate English language revision should be necessary.

Indeed, to facilitate the reader's understanding of the text, it would be better to structure the sentences in the passive form.

Moreover, the Authors should change the citation method of two or more article.

For examples, the text “[1], [2]” should be change in “[1,2]”, please check the Authors guide line.

Corrected by the MDPI service. Certificate added. 

Reviewer 3 Report

Authors present a very interesting work and idea. Metabolic diseases identification and monitoring, preferably with no invasive methods is a ‘hot’ topic and this paper gives a step forward on this.

On the other hand there are many shortbacks in the design of the present study and the way is presented.

Study design and animal selection is unclear and needs to be re-written in detail.

It is not clear how animals were selected. What I understand is that authors went in a farm for a month (July 2023) grossly 8 times (2 days per week) and used the in line “BROLIS HerdLine in-line milk analyzer” to check the F/P ratio in milk. Then authors selected cows with abnormal F/P ratio and clinically examined them. There is not information about:

The number of cows examined, did all the selected cows included in the study?

The selection criteria besides milk F/P. Was that the only criterion for selection?

The lactation stage of the cows. It is presented in the table 1, as DIM, were we see that cows are between 4th and 211th DIM. Especially for metabolic diseases diagnosis’ it is not usual to compare/ group cows of first week of lactation with others at the 7th month!

The parity number. Again from table 1 is assumed that all animals are grouped together parity 1-9, which again is not usual (and useful) in metabolic disease studies

The grouping of animals in the farm. Authors provide the information (L119-120) that all cows are fed the same diet, formulated for 37 kg milk/d per cow. So all cows are one group and fed the same? This makes it difficult for monitoring metabolic diseases. Not irrelevant the cases of SCA had mean DIM the double (almost 100 days) than the other cows.

Inclusion of cows. Are all cows included once? Could a healthy cow classified e.g. as control (H) at one measurement, also classified as SCK e.g. three weeks later? Same cows were examined at each visit? This should be clarified.

So if the study design is to visit a farm, measure the milk F/P ratio in all cows and with that criterion, then examine clinically animals and blood sample them, and classify them in one of 3 groups, there are problems concerning too random animal selection. It is not easy to compare animals with so different DIM, age (animal milk production on inclusion day is nowhere provided), milking group (if different) and extract correct conclusions.

It is also mentioned by authors (L309-311, 348,…).

Suggestion is that the animal selection part in method should be revised with addition of all the above mentioned information and make the study design clear, especial on animal selection and inclusion criteria.

Discussion part needs improvement. It is more a literature parathesis, like a review, and less discussion is implementing the current study results. Also the SCA, acidosis, is a very interesting finding that deserves part in discussion. If the milk F/P ratio can contribute to ruminal acidosis diagnosis is sth that could be stressed accordingly.

 Detailed comments

Key words: in line, innovation are suggested to change 

L128: farms were visited. Only one farm is mentioned, so singular here

L156: this is not clear. Authors visited twice a week the farm, but these samples were taken daily? From which cows? And which of these samples are presented in the study?

L158-162: Again, by which cows? How these samples measured and where are the results presented?

L163-175: as mentioned above cows selection, grouping, farm details, need clarification

L185-186: This NEFA classification (group1-2) is not mentioned anywhere in results or discussion. Were cows classified like this? It could be difficult with so different DIM days.

L279-280: It is not clear what is compared. Nowhere are mentioned primiparous cows in the study, nor BHB is presented on this group

L284: R/B?

L309-311: This is not clear in the current study. How feed restriction was implemanted? Did authors mean a literature refferred study? In any case this discussion makes it more difficult to understand why authors choose to compare NEFAs in a group o animals with DIM variation from 4 to 211

L348-349: correct, but authors designed a study without taken this in to consideration, though it is supposed that knew the cows’ data.

Author Response

Dear Reviewer, 

Authors are very thankful for the comments, which help us to improve the manuscript. All changes proposed have been included in the manuscript and highlighted in yellow and track changes.  

Best Regards, 

Prof. Ramunas Antanaitis 

Question  

Answers  

 Authors present a very interesting work and idea. Metabolic diseases identification and monitoring, preferably with no invasive methods is a ‘hot’ topic and this paper gives a step forward on this.

On the other hand there are many shortbacks in the design of the present study and the way is presented.

Thank you for your positive comment on the article

Study design and animal selection is unclear and needs to be re-written in detail.

We corrected study design section in detail.

It is not clear how animals were selected.

 What I understand is that authors went in a farm for a month (July 2023) grossly 8 times (2 days per week) and used the in line “BROLIS HerdLine in-line milk analyzer” to check the F/P ratio in milk. Then authors selected cows with abnormal F/P ratio and clinically examined them. There is not information about: 

The number of cows examined, did all the selected cows included in the study?

The selection criteria besides milk F/P. Was that the only criterion for selection?

The lactation stage of the cows. It is presented in the table 1, as DIM, were we see that cows are between 4th and 211th DIM. Especially for metabolic diseases diagnosis’ it is not usual to compare/ group cows of first week of lactation with others at the 7th month! 

The parity number. Again from table 1 is assumed that all animals are grouped together parity 1-9, which again is not usual (and useful) in metabolic disease studies

The grouping of animals in the farm. Authors provide the information (L119-120) that all cows are fed the same diet, formulated for 37 kg milk/d per cow. So all cows are one group and fed the same? This makes it difficult for monitoring metabolic diseases. Not irrelevant the cases of SCA had mean DIM the double (almost 100 days) than the other cows.

We added information in the methods section about animals selection and corrected a few mistakes (days in milk, number of lactation).

We added information – “2.2. Experimental Design. Throughout the study period, the farm was visited two times per week on the same days (on Tuesdays and Thursdays) each week. According to the milk fat-to-protein ratio registered by in -line analyzer (Brolis Sensor Technology, Vilnius, Lithuania of 1160 cows, 320 (second and more lactation from the first 5 to 30 days after calving) were selected. Clinical examinations were performed on all 320 cows. According to the results of the clinical examination and blood BHBA from 320 cows for this study, 96 cows were selected: in the SCK group, 62 were selected for SCA, 14, and in the healthy group, 34 cows. The same cows were examined at each visit. 14 cows that at the start of the study were assigned to the healthy group and during the experiment became sick were removed from this study. In this case, the final number in the healthy group was 20.  

From 1160 cows for clinical examination, 320 were selected (second and more lactation from the first 5 to 30 days after calving)). We corrected it to "first 5 to 30 days after calving", because it was a mistake. 

Cows were selected second and more lactation from 5 to 30 days after calving).

Cows were selected second and more lactation

all cows are fed the same diet

Inclusion of cows. Are all cows included once? Could a healthy cow classified e.g. as control (H) at one measurement, also classified as SCK e.g. three weeks later? Same cows were examined at each visit? This should be clarified.

So if the study design is to visit a farm, measure the milk F/P ratio in all cows and with that criterion, then examine clinically animals and blood sample them, and classify them in one of 3 groups, there are problems concerning too random animal selection. It is not easy to compare animals with so different DIM, age (animal milk production on inclusion day is nowhere provided), milking group (if different) and extract correct conclusion

We corrected to – “. Throughout the study period, the farm was visited two times per week on the same days (on Tuesdays and Thursdays) each week. According to the milk fat-to-protein ratio registered by in -line analyzer (Brolis Sensor Technology, Vilnius, Lithuania of 1160 cows, 320 (second and more lactation from the first 5 to 30 days after calving) were selected. Clinical examinations were performed on all 320 cows. According to the results of the clinical examination and blood BHBA from 320 cows for this study, 96 cows were selected: in the SCK group, 62 were selected for SCA, 14, and in the healthy group, 34 cows. The same cows were examined at each visit. 14 cows that at the start of the study were assigned to the healthy group and during the experiment became sick were removed from this study. In this case, the final number in the healthy group was 20

It is also mentioned by authors (L309-311, 348,…).

Suggestion is that the animal selection part in method should be revised with addition of all the above mentioned information and make the study design clear, especial on animal selection and inclusion criteria.

Corrected

Discussion part needs improvement. It is more a literature parathesis, like a review, and less discussion is implementing the current study results. Also the SCA, acidosis, is a very interesting finding that deserves part in discussion. If the milk F/P ratio can contribute to ruminal acidosis diagnosis is sth that could be stressed accordingly.

We corrected whole discussion section.

We added – “Modern dairy farming frequently leads to high milk production, which causes health problems in cows [1]. In this study, changes have been compared in the in-line registered milk fat to protein ratio with the help of BROLIS HerdLine for the assessment of metabolic status in dairy cows

„...The results of the current study are in agreement with the studies conducted by other researchers...“

„...We acknowledge that rumen fermentation involves a delicate balance of volatile fatty acids, including acetic acid and propionate, which play vital roles in the overall metabolism of dairy cows. Accordingly, we recommend future studies investigate the relationship between milk F/P, acetic acid, and propionate acid during rumen fermentation. Based on our findings and those of the literature, we may infer that in-line F/P can be used to identify cows at high risk of subclinical ketosis..“

 ...„Zschiesche et al.[20], found significant associations between milk F/P and ruminal pH. F/P has been identified as a promising indicator in the effort to assess  the SCA situation on a farm [29], [30], [31]. In fact, F/Pis frequently regarded in dairy practice, and several research confirm the notion of F/Pas a good SARA indicator (for example, 8 Holstein Friesian cows yielding 25 kg of milk under trial conditions [32]. Others, on the other hand, did not establish a sufficiently tight link between F/P and ruminal pH (for example, 24 transition Holstein Friesian cows on a real farm [33]Danish Holstein cows with 250 DIM in a trial setting [34], and the limitations of FPR as a SARA indicator are explored. Changes in F/P were dependent on both a decrease in milk fat and an increase in milk protein content. A shift of volatile fatty acids (VFAs) in the rumen with increased propionate and decreased acetate has long been recognized as the cause of milk fat depression as a result of too many highly fermentable carbohydrates and insufficient structural effectiveness in the diet [20]. Sutton [35] proposed that fluctuations in the molar proportions of VFAs in the rumen could explain up to 80% of the variation in milk fat. Furthermore, a decrease in milk fat synthesis due to specific products of ruminal fat biohydrogenation is now thought to be a relatively solid explanation [36]. While the physiological principle underlying the relationship between low ruminal pH and low milk fat is well established, the same cannot be said for milk protein. While the decrease in protozoa commonly seen in SARA diets will increase the efficiency of bacterial growth via reduced predation, a low pH is associated with less efficient bacterial growth in general, which will have the opposite effect [30]. While the physiological basis is unclear, Plaizier et al. [14], who also described an increased milk protein content in experimentally induced SARA, or Mensching et al. [29] confirmed the presence of a negative association between FPR and ruminal pH...”

“…Based on our and the literature results, we can conclude that in-line F/P can be used for the identification of cows with a higher risk of subclinical acidosis…”

“…Toni et al., found that the F/P ratio, is a useful indicator of lipomobilization and the NEB in postpartum cows [56].  Considering the results of these and our studies, we can state that the milk fat-to-protein ratio can serve as an indicator for assessing the metabolic condition of cows…”

„..The liver, being a primary organ involved in ruminant metabolism, exhibits sensitivity to alterations in nutrition. Serum AST and GGT are commonly employed as indicators of hepatic injury caused by metabolic disorders or external stresses. The levels of AST and GGT in the bloodstream are elevated in cases of liver injury, leading to the release of these intracellular enzymes into the serum [58]. Furthermore, it has been established that metabolic stress during the early lactation period in cows has an impact on liver health. This is attributed to the negative energy balance, which triggers heightened lipolysis, resulting in excessive lipid accumulation and the development of liver lesions. Consequently, this process leads to an elevation in liver enzymes [59]. Based on our findings, it has been observed that cows affected by SCA have elevated levels of AST and GGT activities. The results obtained in our study are in agreement with the results obtained by Morar et al., which also showed that the activity of AST and glutamate dehydrogenase (GLDH) was significantly higher in spontaneous subacute ruminal acidosisSARA cows (p < 0.05) than in healthy cows [60]. Elevated levels of AST in the bloodstream are regarded as a highly responsive indicator for the detection of hepatocellular injury, even in cases where the injury is not readily apparent [61]. According to the available scientific data, during the initial phase of lactation, dairy cows demonstrated the highest level of AST activity. Nevertheless, as the duration of lactation progressed, a decrease in the enzymatic activity of this specific enzyme was seen [62]..”

“..Based on the results of our and others studies, we can conclude that the milk fat-protein ratio corresponds to lipolysis and ketogenesis indicators in cow blood and is an important predictor of cow metabolic stress. According to Negussie et al., the determination of the milk fat to protein ratio can be achieved through the regular practice of milk recording [58]. The results of our study highlights the significance of milk as a potentially advantageous medium for the detection of subclinical metabolic disorders, including ketosis and acidosis. The convenience of milk sampling and the ability to conduct whole herd testing during routine recording contribute to its potential as a diagnostic matrix. The assessment of fat to protein ratio is non-invasive and can be conducted during each milking session without inducing further distress to the bovine. To make results more reasonable for future studies, we recommend measuring the in-line milk fat-protein ratio with BHBA, triglycerides, cortisol, high-density lipoproteins, and very low-density lipoproteins [59]. Also, as a weakness of our study, we can declare that the SCA group had a small number of animals because, on this farm during the research period, SCA was not very often  cows health problem. Because of this, in future studies, we recommend investigating changes in the in-line registered milk fat to protein ratio for the assessment of metabolic status in dairy cows with the large number of animals…

Detailed comments

Key words: in line, innovation are suggested to change

We corrected to – “precision dairy farming ; milk composition; biomarker; dairy cattle”

L128: farms were visited. Only one farm is mentioned, so singular here

Line 40 sentence “Throughout the study period farms were visited two times per week on the same days each week” was corrected into “Throughout the study period farm was visited two times per week on the same days each week”

L156: this is not clear. Authors visited twice a week the farm, but these samples were taken daily? From which cows? And which of these samples are presented in the study?

We corrected to – “To determine the greatest BHB feasible, samples were taken during farm visits, at the same time relative to feeding each week on each farm..”

L158-162: Again, by which cows? How these samples measured and where are the results presented?

We added information – “During the farm visit, blood samples from 320 cows were gathered and glucose, β-hydroxybutyrate(BHBA), gamma-glutamyltransferase(GGT), aspartate transaminase(AST), albumins, and nonesterified fatty acids (NEFA) concentrations measured. Concentrations of BHBA and glucose levels were assessed using a capillary blood sample obtained from the ear during a clinical examination” Results are presented in table 1.

L163-175: as mentioned above cows selection, grouping, farm details, need clarification

We added additional section – “2.2. Experimental Design”

L185-186: This NEFA classification (group1-2) is not mentioned anywhere in results or discussion. Were cows classified like this? It could be difficult with so different DIM days.

We deleted this sentence because cows were not classified by NEFA.

L279-280: It is not clear what is compared. Nowhere are mentioned primiparous cows in the study, nor BHB is presented on this group

We deleted this sentence because in this stady were multipatous cows.

L284: R/B?

“R/B” was corrected into “F/P” (Line 289).

L309-311: This is not clear in the current study. How feed restriction was implemanted? Did authors mean a literature refferred study? In any case this discussion makes it more difficult to understand why authors choose to compare NEFAs in a group o animals with DIM variation from 4 to 211

This is part of literature. But we deleted this part about feed restriction.

L348-349: correct, but authors designed a study without taken this in to consideration, though it is supposed that knew the cows’ data.

We added information – “From 1160 cows for clinical examination, 320 were selected (second and more lactation from the first 5 to 30 days after calving.”

Reviewer 4 Report

The data presented in this manuscript is sound and the analysis are well performed. The manuscript assessed cow health by analyzing milk composition data collected by online sensors and found that SCK cows had higher F/P ratios. However, there are some issues in this paper that need to be modified and improved.

Introduction

Line 69. Please give the full name of NEFA the first time it appears in the manuscript.

Blood samples from the ear?

Materials and Methods

Whether the BROLIS HerdLine in-line milk analyzer's measurement accuracy, resolution and measurement time for individual samples can be briefly described in measurements.

Line 154. Please mark whether plasma samples are collected using evacuated tube with anticoagulant or evacuated tube without anticoagulant.

Cows in the SCK group were classified as having an F/P higher than 1.5 mmol/L (Line 163-166), and the criterion in the statistical analyses was F/P >1.2 (Line 185), so please standardize the criteria. However, in statistical analysis section, “F/P <1.2 (risk of SCA), F/P = 1.2 (healthy), and F/P >1.2 (risk with SCK)”? How do authors make this criteria? It is better to make a correlation analyse between F/P ratio and disease phenotype, then to make a conclusion.

The SCK group (n=62) was too different from the healthy group (n=20) and the SCA group (n=14), please explain or harmonize biological replicates.

All statistical significance labeling is wrong in whole manuscript, you should use the word A B C.. method comparing among the three groups, not the asterisk method.

Is it useful to measure ketone bodies, triglycerides (TG), cortisol (Cor), high-density lipoproteins (HDL-C), and very low-density lipoproteins (VLDL-C) in the serum, which can make the results more reasonable.

Positive correlations were identified between blood BHB and NEFA levels (r = 0.321, p < 0.01), as well as a negative association between BHB and glucose concentrations (r = -0.330, p < 0.01).” The correlation coefficients were too low to to be indicative. Thus, the Fig6 and Fig7 are meaningless.     

The pearson correlation coefficient analysis in the statistical analysis should be supplemented with appropriate details.

β-hydroxy-butyrate (BHB)? Generally, we used BHBA for β-hydroxy-butyrate.

It’s similar to having a small laboratory on your dairy farm? What do you mean?

Results

Abbreviations for items in Table 1 and Table 2 are mixed with throughout, please harmonize. And label the source, e.g. GGT (from serum).

In Table 2, “**. Correlation is significant at the 0.01 level (2-tailed)” and “*. Correlation is significant at the 0.05 level (2-tailed)” are more appropriately placed in the notes below.

Discussion

Excessive references are cited in the discussion, and the discussion should adequately address the author's own views.

The manuscript are well-prepared for English language.

Author Response

Dear Reviewer, 

Authors are very thankful for the comments, which help us to improve the manuscript. All changes proposed have been included in the manuscript and highlighted in yellow and track changes.  

Best Regards, 

Prof. Ramunas Antanaitis 

Question  

Answers  

 The data presented in this manuscript is sound and the analysis are well performed. The manuscript assessed cow health by analyzing milk composition data collected by online sensors and found that SCK cows had higher F/P ratios. However, there are some issues in this paper that need to be modified and improved.

Thank you for your positive comment and your suggestions.

Introduction

Line 69. Please give the full name of NEFA the first time it appears in the manuscript.

Blood samples from the ear?

We added – „...nonesterified fatty acids (NEFA)…” Also, we added explanation of all abbreviations – “During the farm visit, blood samples from 320 cows were gathered and glucose, β-hydroxybutyrate(BHBA), gamma-glutamyltransferase(GGT), aspartate transaminase(AST), albumins, and nonesterified fatty acids (NEFA) concentrations measured”

We corrected – “During the farm visit, blood samples from 320 cows were gathered and glucose, β-hydroxybutyrate(BHBA), gamma-glutamyltransferase(GGT), aspartate transaminase(AST), albumins, and nonesterified fatty acids (NEFA) concentrations measured. Concentrations of BHBA and glucose levels were assessed using a capillary blood sample obtained from the ear during a clinical examination. Capillary blood samples were taken at the ear's. Cleaning and puncturing the skin of the left or right ear was the first step in the sampling technique. If the blood volume was insufficient for the measurement, capillary bleeding was induced by gently pressing the ear skin. If the blood volume retrieved was still insufficient for a meaningful measurement, the ear was punctured again. The front edge of the test strip was dipped immediately into the drop of blood after being inserted into the handheld device [21]. The Medi Sense and Free Style Optium H systems (Abbott, Great Britain) were utilized to measure the amounts of plasma BHBA and glucose.

At the same time, blood samples from the jugular vein were taken using an evacuated tube with no anticoagulant (BD Vacutainer®, Eysin, Switzerland). The blood samples…”

Whether the BROLIS HerdLine in-line milk analyzer's measurement accuracy, resolution and measurement time for individual samples can be briefly described in measurements.

We added information   and Figure 2– „During each milking, the analyzer measures the composition of each cow's milk continuously throughout the milking, every 5 seconds. Fat, protein, and lactose concentration dynamics are averaged with weights based on milk flow to obtain single values to represent whole milking. Milk analyzer accuracy was evaluated in the Eurofins lab with resulting values of root mean square error of prediction (RMSEP) of 0.21% for fat, 0.19% for protein, and 0.19% for lactose. This "mini-spectroscope" is mounted in the milking stalls or milking robot in the milk line and does not require any additional reagents or maintenance(Figure 2)

Line 154. Please mark whether plasma samples are collected using evacuated tube with anticoagulant or evacuated tube without anticoagulan

We corrected information to – „….vein were taken using an evacuated tube with no anticoagulant (BD Vacutainer®, Eysin, Switzerland)”

Cows in the SCK group were classified as having an F/P higher than 1.5 mmol/L (Line 163-166), and the criterion in the statistical analyses was F/P >1.2 (Line 185), so please standardize the criteria. However, in statistical analysis section, “F/P <1.2 (risk of SCA), F/P = 1.2 (healthy), and F/P >1.2 (risk with SCK)”? How do authors make this criteria? It is better to make a correlation analyse between F/P ratio and disease phenotype, then to make a conclusion.

We corrected to – „Cows data were classified according to their milk fat and protein ratios: F/P <1.2 (risk of SCA), F/P = 1.2 (healthy), and F/P >1.5 (risk with SCK) [22]”

The SCK group (n=62) was too different from the healthy group (n=20) and the SCA group (n=14), please explain or harmonize biological replicates.

We added information – „Clinical examinations were performed on all 320 cows. According to the results of the clinical examination and blood BHBA from 320 cows for this study, 96 cows were selected: in the SCK group, 62 were selected for SCA, 14, and in the healthy group, 34 cows. The same cows were examined at each visit. 14 cows that at the start of the study were assigned to the healthy group and during the experiment became sick were removed from this study. In this case, the final number in the healthy group was 20

Also, small number of cows in SCA was because  that during exeperiment in this farm SCA was not very often problem. In methods section we added – “Clinical examinations were performed on all 320 cows. According to the results of the clinical examination and blood BHBA from 320 cows for this study, 96 cows were selected: in the SCK group, 62 were selected for SCA, 14, and in the healthy group, 34 cows. The same cows were examined at each visit. 14 cows that at the start of the study were assigned to the healthy group and during the experiment became sick were removed from this study. In this case, the final number in the healthy group was 20

According to this, in discussion section we added – “Also, as a weakness of our study, we can declare that the SCA group had a small number of animals because, on this farm during the research period, SCA was not very often  cows health problem. Because of this, in future studies, we recommend investigating changes in the in-line registered milk fat to protein ratio for the assessment of metabolic status in dairy cows with the large number of animals”

All statistical significance labeling is wrong in whole manuscript, you should use the word ‘A B C..’ method comparing among the three groups, not the asterisk method.

We corrected in whole manuscript

Is it useful to measure ketone bodies, triglycerides (TG), cortisol (Cor), high-density lipoproteins (HDL-C), and very low-density lipoproteins (VLDL-C) in the serum, which can make the results more reasonable.

We added information – „To make results more reasonable for future studies, we recommend measuring the in-line milk fat-protein ratio with BHB, triglycerides, cortisol, high-density lipoproteins, and very low-density lipoproteins [58]”

“Positive correlations were identified between blood BHB and NEFA levels (r = 0.321, p < 0.01), as well as a negative association between BHB and glucose concentrations (r = -0.330, p < 0.01).” The correlation coefficients were too low to to be indicative. Thus, the Fig6 and Fig7 are meaningless.  

We deleted Fig 6 A and C and Fig 7 A, B and C.

The pearson correlation coefficient analysis in the statistical analysis should be supplemented with appropriate details.

We added information – „The Pearson correlation coefficient was determined to define the linear relationship between the variables under consideration. The statistical link between milk in-line fat to protein ratio and blood biochemical markers was determined using a linear regression equation. It was considered dependable (p = 0.05) if the probability was less than 0.05”

β-hydroxy-butyrate (BHB)? Generally, we used BHBA for β-hydroxy-butyrate.

We corrected to – „β-hydroxybutyrate (BHBA)” in whole manuscript.

It’s similar to having a small laboratory on your dairy farm? What do you mean?

We deleted this sentence.

Abbreviations for items in Table 1 and Table 2 are mixed with throughout, please harmonize. And label the source, e.g. GGT (from serum).

We harmonize abbreviations and added the source.

In Table 2, “**. Correlation is significant at the 0.01 level (2-tailed)” and “*. Correlation is significant at the 0.05 level (2-tailed)” are more appropriately placed in the notes below.

Corrected to – „* Correlation is significant at the 0.05 level (2-tailed); **Correlation is significant at the 0.01 level (2-tailed);BHB­­A- β-hydroxybutyrate; AST-aspartate transaminase; GGT- gamma-glutamyltransferase; NEFA-nonesterified fatty acids, DIM–days in milk”

Discussion

Excessive references are cited in the discussion, and the discussion should adequately address the author's own views.

We corrected the discussion section and added our opinion after each paragraph. 

Also, in conclusion section we added – „We think that comparing the metabolic data of blood and milk reveals that the milk fat-to-protein ratio can serve as an indicator for assessing the metabolic condition of cows. Milk characteristics play a pivotal role in predicting cow metabolic stress, as they correspond to indicators of lipolysis and ketogenesis in cows. Furthermore, the non-invasive methods of milk sample collection make it a suitable method for routine assessment of metabolic status. We believe that this could be a beneficial complement to dairy farm herd health programs”

The manuscript are well-prepared for English language.

Thank you for your positive comment.

Round 2

Reviewer 1 Report

The revision of manuscript number 2584422 received from ANIMALS, with title: “In-line Registered Milk Fat/Protein Ratio For The Assessment Of Metabolic Status In Dairy Cows” has been evaluated.

The paper has been radically improved; however, some critical issues remain that need to be clarified and/or implemented:

Row               comment

39                  remove  “mmol/L”

124                remove  “%”

142                remove  “mmol/L”

177-179        this sentence is the same than that in rows 162-164?

184                authors have sampled 320 cows then selected 62 for SCK, 14 for SKA, 20 for H groups. All the others?  Why were not suitable for any of the groups?

187-209        authors should explain why have used two different sides to sample blood for the same parameter

216                HG = H (?)

Table 1          when are added letters to indicate statistical significance all the means of the parameter should have a proper letter that indicate the relationship with other, as well if in the table are reported uppercase also in the footnote must be uppercase, not lowercase as in row 231. Moreover, in row 232 should be removed “* p < 0.05, ** p < 0.01;” as not present in the Table.

Results & Discussion                   in these two sessions the p values are reported for several parameters, but it is never reported respect which mean, as in the design there are 3 groups this is compulsory

References   in the paper appear citation till 62 but the list of papers is stopped at 54. Please check

                       Check format of reference n. 52

Author Response

Dear Reviewer, 

Authors are very thankful for the comments, which help us to improve the manuscript. All changes proposed have been included in the manuscript and highlighted in yellow and track changes.  

Best Regards, 

Prof. Ramunas Antanaitis 

NB: Q – your question or comment; A – answer.

Q: 39                  remove  “mmol/L”

A: Deleted

Q: 124                remove  “%”

A: Deleted

Q: 142                remove  “mmol/L”

A: Deleted

Q: 177-179        this sentence is the same than that in rows 162-164?

A: We deleted this sentence

Q: 184                authors have sampled 320 cows then selected 62 for SCK, 14 for SKA, 20 for H groups. All the others?  Why were not suitable for any of the groups?

A: They were not suitable because they don’t fit in these categories – “second and subsequent lactation from the first 5 to 30 days after calving)” L126-127.

Q: 187-209        authors should explain why have used two different sides to sample blood for the same parameter

A: We don’t used two different sides to sample for the same parameter. Different sides were used for different parameters.  L 187-188 – “The concentrations of BHBA and glucose levels were assessed using capillary blood sam-ples obtained from the ear during each clinical examination” L. 196-205 – “At the same time, blood samples from the jugular vein were taken using an evacuat-ed tube with no anticoagulant (BD Vacutainer®, Eysin, Switzerland). The blood samples were centrifuged for 10-15 minutes at 3500 RPM. The samples were delivered to the Lith-uanian University of Health Sciences Veterinary Academy's Large Animal Clinic's Labor-atory of Clinical Tests. Blood serum was tested using a Hitachi 705 analyzer (Hitachi, To-kyo, Japan) and DiaSys reagents (Diagnostic Systems GmbH, Berlin, Germany) to deter-mine the activities of GGT and AST and the concentrations of albumins. The NEFA sam-ples were evaluated using an automated wet chemistry analyzer (Rx Daytona, Randox Laboratories Ltd., London, UK) and Rx Daytona reagents (Randox Laboratories Ltd., London, UK)”

Q: 216                HG = H (?)

A: Corrected to “H”

Q: Table 1          when are added letters to indicate statistical significance all the means of the parameter should have a proper letter that indicate the relationship with other, as well if in the table are reported uppercase also in the footnote must be uppercase, not lowercase as in row 231. Moreover, in row 232 should be removed “* p < 0.05, ** p < 0.01;” as not present in the Table.’

A: We corrected Table 1 according to your comments L234-235.

A: Corrected to – “The letters A,  B and  C  show statistically significant differences in the mean values”

A: “* p < 0.05, ** p < 0.01” – deleted.

Q: Results & Discussion                   in these two sessions the p values are reported for several parameters, but it is never reported respect which mean, as in the design there are 3 groups this is compulsory.

A: We corrected –

 L 240-243 “We found a higher level of milk fat-to-protein ratio in cows with SCK compared with SCA and H groups. The average in-line F/P ratio of SCK cows was significantly higher than that of SCA and H cows (p < 0.01). The average in-line F/P ratio of SCK cows was 1.66 (± 0.29) and of SCA cows was 0.93 (± 0.1), and that of healthy cows was 1.22”

L253-254 “According to our results, we found a significantly higher AST activity of 26.66% in cows with SCA compared with the healthy group (p < 0.05)”

L266- 267 “We obtained a significantly higher GGT activity (p < 0.05) by 12.72% in cows with SCA compared to cows of the healthy group (p < 0.05)”

L278-279 “In cows with SCK, a significantly higher NEFA concentration of 40.38% was detected compared to healthy cows (p < 0.001)”

Q: References   in the paper appear citation till 62 but the list of papers is stopped at 54. Please check

 Check format of reference n. 52

A: Corrected.